# Novel approach for predicting fake news stance detection using large word embedding blending and customized CNN model

**Abdulaziz Altamimi** [ORCID] *

Department College of Computer Science and Engineering, University of Hafr Al-Batin, Hafar Al-Batin, Saudi Arabia

* Dr.Altamimi@uhb.edu.sa

## Abstract

The proliferation of fake news is one of the major problems that causes personal and societal harm. In today's fast-paced digital age, misinformation spreads rapidly, often leaving individuals without the time to verify the authenticity of the information. This can cause irreparable damage to personal reputations and organizational credibility. Thus, instigated by the quintessential necessity, there is a dire need to construct a framework for the automatic detection and identification of fake news at its inception. This research presents a novel approach that leverages a combination of three popular word embeddings (FastText, FastText-Subword, and GloVe) integrated with a customized convolutional neural-network (CNN) to classify fake news accurately. The proposed model was tested against the Fake News Challenge dataset. Hundreds of word vector features were generated from the combined embedding and then managed with PCA and significant features were extracted. The proposed model gives an accuracy of 94.58%, 95.35% precision, 97.29% recall, and an F1 score of 96.11%. The proposed framework's robustness is demonstrated when compared with other machine, deep, and ensemble learning approaches, showing superior performance. Furthermore, the effectiveness of the model is validated on an independent Arabic Fake News dataset.

## 1 Introduction

Digital media has immensely contributed to the enrichment of today's society in a way that it has made information spread easier, improved social engagements, and increased work efficiency. The negative is that digital media allows fake news [1, 2], intentionally fabricated information. The detrimental effects of misinformation [3] are widespread across all societal segments. Examples like the Pizzagate conspiracy [4] and mob lynchings in India [5] illustrate the severe consequences of disseminating false information. Such misinformation has altered health behavior intentions, contributed to vaccine hesitancy, and resulted in significant economic damages [6]. Given the enormous daily output of digital content, it is unfeasible for humans alone to fact-check and identify false news. Thus, automated methods are essential for the prompt detection and mitigation of fake news.

**Data Availability Statement:** The dataset utilized in this research work are available at these links: 1. https://www.kaggle.com/datasets/mohamedabdelaleem007/labeledarticles 2. https://

www.kaggle.com/datasets/abhinavkrjha/fake-news-challenge.

**Funding:** The author(s) received no specific funding for this work.

**Competing interests:** The authors have declared that no competing interests exist.

This has been highly proliferated, and the same has become a concern for researchers and technology companies to find practical solutions. One of the most promising ways to do this is through machine learning (ML) techniques for fake news spread detection and prevention [7]. ML refers to a subfield system of artificial intelligence that involves using statistical models and algorithms to let computers learn from data and make predictions without being explicitly programmed. The fake news detection (FND) approach is thus of necessity as a counter-measure. Fake news identification on social networks can be inefficient by hand since it requires much labor with relatively poor results. Researchers have established semantic inconsistencies as important marks to confirm the validity of news content on social networks for automated fake news identification. In the past, natural-language-processing methods have been adopted to detect fake information. Deep learning (DL) techniques like neural networks find a place in some of the phony news-detecting tools designed lately.

To be more precise, text-content-based techniques [8, 9] usually take features out of news articles' headlines, phrases, and writing styles. Nevertheless, the majority of these techniques overlook the relationships between sentences, producing less-than-ideal detection outcomes. On the other hand, data mining technologies and user social network data are employed by social network-based detection algorithms [10, 11] to identify fake news. In real-world scenarios, social media platforms enforce strict privacy regulations on user data. Additionally, the lack of availability of specific fake news datasets due to privacy issues can hamper the effective training of detection models. Consequently, implementing detection methods that rely on social network data can be challenging. Researchers have proposed the use of graph neural networks to create detection methods. These graph-based detection strategies [12] produce document graphs using input data. The effectiveness of detection is heavily influenced by the quality of these document graphs' construction. Unfortunately, creating graphs is a difficult and time-consuming procedure that limits the effectiveness and adaptability of various detection techniques.

The fundamental drawback of all the previously stated models is the character limit constraints that some social media platforms, like Twitter, impose on posts, which are usually composed of brief words [13]. These concise text snippets often lack adequate information to authenticate the news. Consequently, individuals are easily deceived by these short and decontextualized messages, accepting and sharing them without thoroughly checking their authenticity. In contrast, a news article offers a more comprehensive account than merely the headline or a tweet related to the news. However, few individuals bother to search for or read the whole article to verify the details [14]. This study employs ML models to classify fake news stances on social media accurately. The primary contribution of the study is;

- An innovative approach for predicting fake news stances is proposed that consists of blending three popular word embeddings(FastText, FastText-Subword, and GloVe) with a customized CNN model.

- Several ML models, including Stochastic-Gradient-Descent(SGD), Random Forest(RF), Logistic Regression(LR), and Voting Classifier(VC), are used to compare the performance of the proposed framework. The effectiveness of the suggested method is then contrasted with the long short-term memory(LSTM) DL model.

- The strength of the proposed framework is checked utilizing cross-validation techniques and by comparing it with previously published research works.

- The reliability of the proposed framework is checked by testing it on an independent dataset of Arabic Fake News detection.

The rest of the paper is structured as Section 2 discusses some closely related published research work in the field of social media-focused fake news stance detection. After that, it describes the research design and the use of ML and DL models in the study in Section 3. Next, experimental results are discussed in Section 4. Section 4 also analyzes the comparison of the proposed framework with previously published research works. Lastly, we wrap up our study and make recommendations for future research possibilities.

## 2 Related work

The proliferation of fake news has a major detrimental effect on individuals and society due to the growth of intelligent media and the information epidemic. Substance detection, fake news stance, and topic detection are the several sub-tasks that makeup FND from the standpoint of certain research projects. Stance identification is an essential subtask that helps extract authenticity cues for detecting fake news. In particular, posture recognition uses natural language processing technologies to evaluate how consistently stance expressions appear in news texts.

To detect fake news, Rezaei et al. [15] used ML techniques such as passive-aggressive(PA), SVM, and NB. But depending just on straightforward classification techniques might not provide the best results. The detection of fake news has been greatly enhanced by combining ML approaches with text-based processing methods. There are difficulties because there aren't enough distinct corpora available for training and differentiation. With accuracy levels as high as 96% their method produced superior findings through experimental analysis on datasets that were made available to the public. The outcomes of the Stage 1 (FNC-1) stance identification test of the Fake News test were enhanced by Valeriya Slovikovskaya [16]. The effectiveness of their work has increased due to the ability of huge language models built on transformer architecture to generalize. The BERT phrase embedding of input sequences was utilized by the author of this paper as a model feature. The BERT, RoBERTa, and XLNet transformers were fine-tuned on the FNC dataset, yielding reliable results with the FNC-1 challenge.

In this line, Turki Aljrees et al. [17] used PCA and chi-square to reduce the number of feature dimensions, as these techniques go with FakeNET hybrid neural network architecture. It was initially developed to combine LSTM and CNN. The reason for using PCA and Chi-square techniques in this research is that it reduces computing complexity; better performance is achieved by using the appropriate feature vectors. It is observed from the experimental results that the accuracy of the proposed PCA-based method is 0.908. To increase the credibility of news in online social networks, Zhibo Zhou et al. [18] developed a deep, innovative fake news stance detection model called the Adversarial-Pseudo-Siamese-Network(APSN) model. The experimental results showed improvements over many competitive benchmarks taken for testing purposes. The maximum score for the proposed model was 93.40.

Rai et al. [19] proposed some techniques based on DL models and natural language analysis. They used a hybrid technique with an attention-based mechanism and a CNN to draw attention to the critical portions of textual data. Due to attention, the CNN models are enabled to focus on particular segments of input data when making predictions or producing outputs. This procedure gives distinct items in a sequence of varying degrees of importance, which helps the model understand the relevant context for improved understanding and generation skills. Techniques based on DL models and natural language analysis were proposed by Rai et al. [19]. They used a hybrid technique that combined an attention-based mechanism with CNN to draw attention to the important portions of textual data. CNN models can focus on particular input data segments for prediction and output production because they use attention. This procedure gives distinct items in a sequence of varying degrees of importance, which helps the model understand the relevant context for improved understanding and

generation skills. CNN-based models have gained popularity because they converge to nearly optimal solutions with low computational complexity, but they are unable to understand past and future dependencies from a text. Three publicly accessible datasets WELFake, FakeNews-Net, and FakeNewsPrediction are used by Ehtesham Hashmi et al. [20] to present a reliable method for FND. They combined a variety of DL and ML techniques with FastText word embeddings, honing these algorithms through hyperparameter optimization and regularization to reduce overfitting and advance model generalization. Across all datasets, a hybrid model that combined LSTM and CNN and was enhanced with FastText embeddings demonstrated superior classification performance compared to other methods, having F1 scores of 0.99, 0.97, and 0.99, respectively, and accuracy. The study of Deepali Goyal Dev et al. [21] focuses on classifying and discovering fake news in two steps. At first, they studied the basis of fake news on social media sites. Afterward, using supervised ML algorithms, they evaluate current FND tactics during the discovery phase. Unlike conventional text analysis techniques, this study employs voice and text features as well as analytical representations to identify fake news. Impressively, the hybrid CNN/LSTM technique yielded an accuracy rate of 0.98. A 'ConFake' method was presented by Jain et al. [22] to automatically detect fake news using content-based criteria. Features derived from news stories' textual content that are both content-based and word vector. ML classifiers were trained with a combination of these features. To validate the experimental results, they ran all of the tests on five datasets that were made available to the public and one purposely constructed dataset called ConFake. Comparing the proposed model to other state-of-the-art models, it achieved a maximum accuracy of 97.31%.

Bengali is a low-resource language; Nipa Rani Das et al. [23] worked on it for FND. In their study, DL and ML models have been used to classify phony news. Their study results show that among such learning algorithms, the LSTM performed the best and gave an accuracy score of 96.14% during fake news recognition. Aytug Onan and Mansur Alp Tocoglu addressed [24] the inadequacies of previous models to classify information as genuine or fake. They proposed a methodology for estimating the credibility of facts using the stacked LSTM model. Their approach utilized multiple word embeddings and hand-crafted features. The results from the experimentation reveal that MLP performed better than baseline ML models in accuracy for GloVe vectorized triples and count vectorized. Furthermore, an approach was given by the authors for concurrent extraction of named entity tags and triples, which served as an additional feature for model training. Table 1 summarizes the related work.

## 3 Materials and methodology

This section will express all the details about the developed fake news identification framework including dataset statistics. The proposed framework architectural methodology flow for FND is shown in Fig 2.

### 3.1 Dataset-1

To promote the creation of automated methods for detecting fake news using AI and ML researchers [25] established the inaugural Fake News Challenge(FNC) to assess news sources' coverage of specific topics. Approximately fifty teams from the business and academic sectors took part in this competition. Finding a news article's position about a title is the aim of the FNC-1 challenge. Four different article stances are possible. It may debate the same subject as the headline, argue with it, or be irrelevant. Their official website provides details about the FNC-1 task, its guidelines, the dataset, and the evaluation measures.

On the official website, the benchmark dataset for Fake News Challenges was made accessible. With 2, 587 article bodies and 75, 385 tagged instances, the FNC dataset is nearly 300

**Table 1. Summarization of literature review.**

| Ref | Year | Classifiers | Dataset | Performance | Limitation |
|---|---|---|---|---|---|
| [15] | 2022 | RF, SVM, DT, LGBM, XGBoost, and stacked ensemble model | Politifact+ Snopes+ Truthorfiction | 96.24% Stacked ensemble model on Politifact dataset | Not applied to low-resource language |
| [16] | 2019 | featMLP, BERT, XLNet, Roberta | FNC-1 | 93.19% RoBERTA | Linear models and machine learning model not used |
| [17] | 2023 | BERT, XLNet, Roberta and FakeNet | FNC-1 | 97.8% FakeNet with PCA features | This research is limited to the English language. |
| [18] | 2023 | GBC, CNN+GBDT, CS, ES+LSTM, ES+LSTM +AT | FNC-1 | 93.4% ES+LSTM+AT | features in the dataset not co-related |
| [19] | 2022 | TCNN-URG, LIWC, CSI, HAN, SAFE, BERT, BERT+LSTM | Politifact, gossip cop | 88.75% BERT+LSTM | ability to parallelize training and inference |
| [20] | 2024 | CNN,-LSTM, BiLSTM-GRU, LSTM, BiLSTM, GRU, BiGRU, CATBoost, ADA, RF, SVM, LR, DT | WELFake, FakeNewsNet, and FakeNewsPrediction | 99% CNN-LSTM | All the dataset belongs to English language |
| [21] | 2024 | ADA, LR, ANN, CNN+LSTM | Kaggle | 98% CNN-LSTM | model is not cross-language dependent |
| [22] | 2024 | KNN, RF, GNB, LR, Bagging, ADA, SVM | ConFake dataset | 97.31 RF | research has been conducted using customized datasets |
| [23] | 2024 | ETC, KNN, RF, MNB, DT, XGBoost | BANS dataset | 96.14% LSTM | Model tested only on low resource language |
| [24] | 2021 | CNN, RNN, LSTM, GRU, BiLSTM, Stacked LSTM | Sarcasm version 1, Sarcasm version 2, The News headline dataset | 95.30% Stacked LSTM | Model not test on any low resource language |

headlines long. For each allegation, there are five to twenty news items. According to Table 2, out of these headlines, 7.4% are accepted, 2.0% are disputed, 17.7% are up for debate, and 72.8% are unrelated. The statements made about the article bodies have precise labels. The following are the labels' specifics:

1. Agree: The headline and text of the article are related.

2. Disagree: The article body and headline are unrelated.

3. Discuss: The headline and article body somewhat match, therefore it can be interpreted as neutral.

4. Unrelated: The subject matter covered in the headline and the body differs greatly.

Training data was made up of 49,972 instances for training whereas the testing data entailed 25,413 instances. Taking into account the FNC challenge guidelines, the data was divided into the training and testing sets. The training data contained 1,648 headlines and 1,683 article bodies, while the test data contained around 880 headlines and 904 article bodies.

### 3.2 Dataset-2

Numerous low-resource datasets can be used to spot fake information. The Arabic Fake News Dataset (AFND), which is accessible to the public on Kaggle, is used in this study to assess how

**Table 2. Statistics of the dataset.**

| Dataset | Tokens | Headline | Instance | Agree(A) | Disagree(D) | Discuss(Di) | Unrelated(U) |
|---|---|---|---|---|---|---|---|
| FNC | 372 | 2587 | 75383 | 7.40% | 2% | 17.70% | 72.80% |

well the proposed system works [26]. Public Arabic news stories are included in this benchmark collection. It has 606912 news items total, collected from 134 Arabic news websites that are accessible to the public. "Misbar", a widely accessible Arabic news fact-checking platform, classifies the articles into three categories: credible, not credible, and uncertain.

### 3.3 Proposed research methodology

**3.3.1 Data preprocessing.**   Data preprocessing is a process aimed at converting redundant, missing, excessive, and inconsistent data into a form amenable for model training. This phase involves several steps to improve the usability of the raw data and boost model efficiency. Python libraries such as NLTK and Keras have been utilized to aid in preprocessing tasks like stopword elimination, converting text to lowercase, and tokenization.

**3.3.2 Word embedding techniques.**   In natural language processing, one important step is to convert text data into vectors so that machine learning and deep learning models can be trained. Word embedding methods have gained popularity in recent years for their use in predictive models. In a study on identifying fake news, three word embedding techniques were used: fastText, fastText-subword, and GloVe. These techniques play a crucial role in the process of natural language processing.

**3.3.3 fastText.**   Word embedding in vector form has been applied to different tasks within natural language processing (NLP). In general, word embedding that has been pre-trained on sources like Wikipedia and Google News predicts word context without supervision. They believe that the words are in proximity and share similar contexts [27, 28]. The fastText method was developed by Facebook's FAIR Research Lab. FastText should be considered for vector representation because it has a word difficulty detection tool. Moreover, the system uses morphological information and vectorization from 2 million words, with 600 billion tokens in 300 dimensions. This ability allows for enhancing text classification results by effectively generalizing them. The fastText word embedding vectors are created by adding n-grams together, allowing for the creation of vectors for words that are not well-known.

**3.3.4 fastText subword.**   The use of fastText subwords in training benefits from the sharing of common word roots. With 300 dimensions and 2 million word vectors (600 billion tokens), it incorporates subword information during training [29, 30]. Millions of vectors trained on common tokens or roots make up the fastText subword. It offers support by dissecting words into smaller parts or combining smaller parts into a single word. For example, "for" and "give" can be expressed independently and then joined to obtain dictionary-level representation by using the combination "forgive". Character-level embedding is beneficial for accurately representing misspelled and slang terms.

**3.3.5 GloVe.**   This word embedding technique creates vectors using an unsupervised learning algorithm. GloVe contains 840 billion crawl words and has 300 dimensions. As its name suggests, It records the corpus's global and local characteristics [31, 32]. It is context-independent and organizes words based on their semantic similarity, displaying the linear substructure of the word vector space. The worldwide lexicon of word co-occurrence is used to train it. GloVe is a logarithmic bilinear model that makes inferences about the meaning of words based on how likely it is that they will appear together. The factorization method creates a word context matrix.

**3.3.6 Combining features.**   This research work combines three popular word-embedding techniques(fastText, fastText-subword, and GloVe). These word-embedding techniques are independently taught to assist the DL models in improving predictive results. A DL model is trained using three word-embedding approaches separately at first, and subsequently by combining them at a ratio of 34% for the fastText subword and 33% for fastText and Glove. The

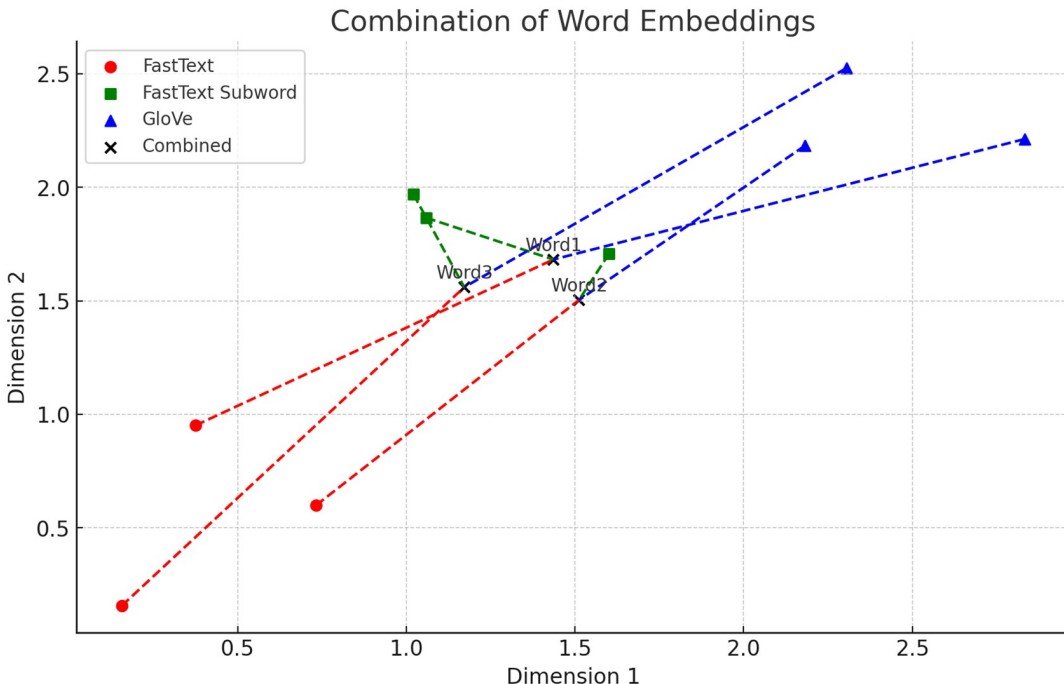

**Fig 1. Illustration of word embedding fusion.**

illustration of combining features is shown in Fig 1. In the diagram, Red circles (o) represent the FastText embeddings, Green squares (s) represent the FastText-Subword embeddings, Blue triangles (∧) represent the GloVe embeddings, and Black crosses (x) represent the combined embeddings. The steps that are involved in combining features are:

1. **Load pre-trained embeddings**: Ensure all embeddings are loaded correctly.

2. **Normalize embeddings**: Make sure all embeddings have the same dimensions.

3. **Combine embeddings**: Compute the weighted average of the embeddings for each word.

4. **Handle missing words**: Provide a default embedding for words not present in all embeddings.

5. **Save the combined embeddings**: Store the final embeddings for future use.

```
combined_embeddings = {}
alpha, beta, gamma = 0.33, 0.33, 0.34

For word in fasttext_embeddings:
    if word in fasttext_subword_embeddings and word in glove_embed
    dings:
        combined_embedding = (alpha * fasttext_embeddings[word] +
                              beta * fasttext_subword_embeddings[word] +
                              gamma * glove_embeddings[word])
        combined_embeddings[word] = combined_embedding
    # Handle words not present in all embeddings
```

```
elif word in fasttext_subword_embeddings:
    combined_embedding = (alpha * fasttext_embeddings[word] +
                          beta * fasttext_subword_embeddings[word] +
                          gamma * np.zeros(embedding_dim))
    combined_embeddings[word] = combined_embedding
elif word in glove_embeddings:
    combined_embedding = (alpha * fasttext_embeddings[word] +
                          beta * np.zeros(embedding_dim) +
                          gamma * glove_embeddings[word])
    combined_embeddings[word] = combined_embedding
else:
    combined_embedding = alpha * fasttext_embeddings[word] + beta
    * np.zeros(embedding_dim) + gamma * np.zeros(embedding_dim)
    combined_embeddings[word] = combined_embedding

# Repeat similar steps for words in fasttext_subword_embeddings
and glove_embeddings that were not in fasttext_embeddings
```

**3.3.7 Dimensionality reduction.** Using various word embedding methods together results in a higher quantity of features and repetitions, which adds extra workload to the classifier training process. Using a suitable method for selecting features can address this issue [33]. PCA is a popular tool for classification jobs, as it reduces the number of characteristics by using a linear transformation. Following PCA reduction, the dataset contains features that were present in the original data, and the covariance matrix is used to determine the principal component. Using a pool of 3400 features, PCA is used in this study to select significantly important 1500 features for the experiments.

**3.3.8 Modeling methods.** Big data presents a variety of computational issues that CNN overcomes by using nonlinear-activation, dropout, pooling-layers, and filters [34, 35] to obtain complex information [36, 37]. This model training is more effective because it is completed end-to-end. Semantic information is encoded at the end of the model using fully connected layers. It is a network that advances by filtering the layer's output before identifying features. Convolutional layers, pooling layers, a flattened-layer, dropout, activation functions, and the dense layer are the main components of the CNN model. Convolutional layers first extract features, which are then sent to fully connected layers for further analysis. To minimize overfitting risks, the number of features detected by convolutional layers is curtailed by the pooling layer. Pooling can be executed using either a max or average layer; with max-pooling selecting different features compared to average pooling. The flattened layer converts the data into an array prior to its passage to the fully connected layer. The rectified-linear-unit(ReLU) activation function is utilized in model development:

$$y = max(0, i) \tag{1}$$

'$y$' is the output activated, while '$i$' is the input given. Convolutional layers are utilized to find the patterns in the dataset with the help of weights applied to kernels during training. Now 1D-CNN model shows great results with text classifications as in our case. The binary

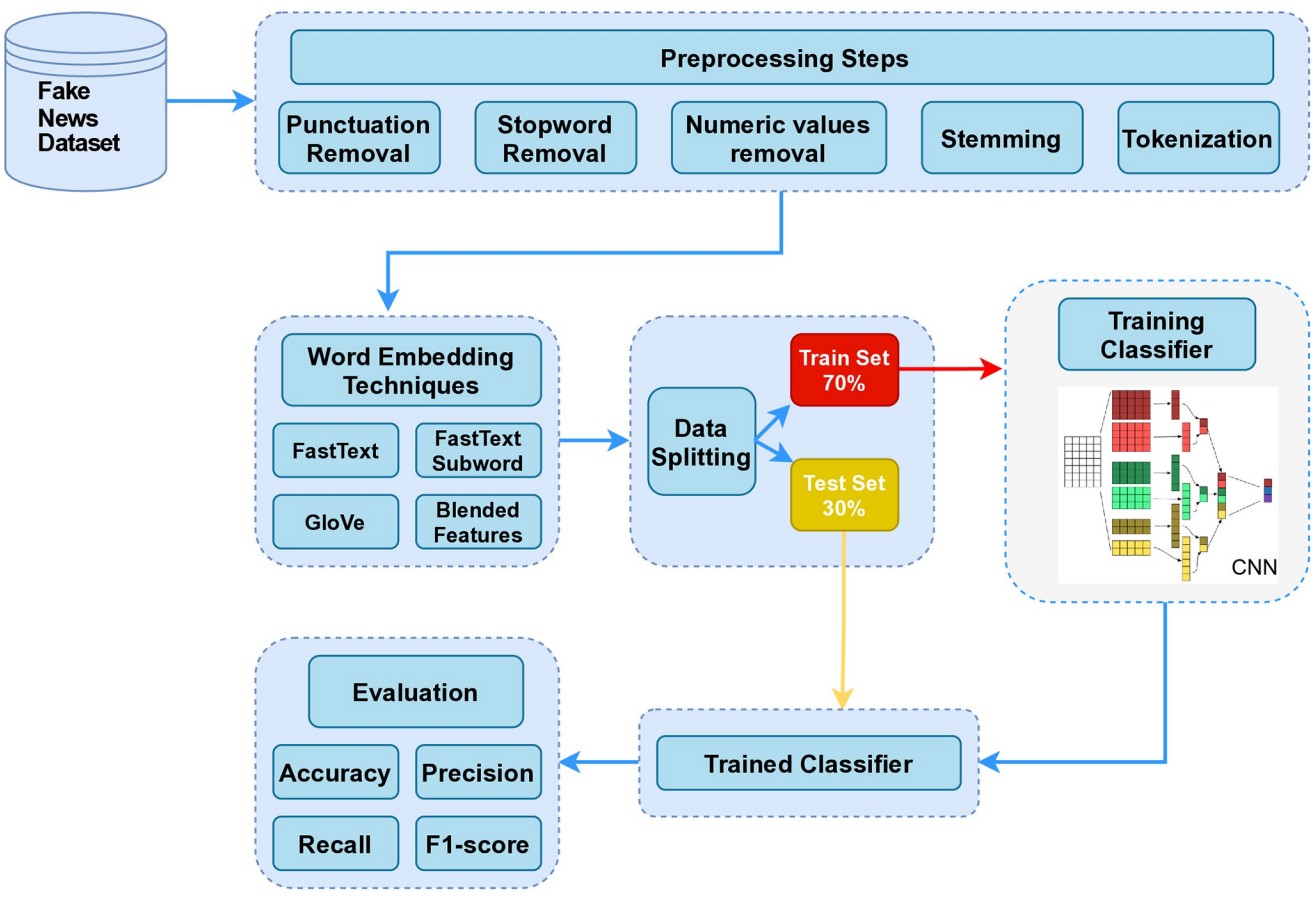

**Fig 2. Proposed fake news stance detection architecture.**

cross-entropy is utilized to calculate loss during training. Its formula is:

$$crossentropy = -(i \times log[(p) + (1 - i)log(2 - p)]) \tag{2}$$

'$i$' denotes class labels, log refers to the natural log, and p signifies the predicted probability. Given that CNN is a variant of the back propagation technique, the output's error function is sigmoid [38]. The CNN model produces three neurons as output for every instance of the target class. CNN is considered a strong model for classifying tasks within the medical domain. Many researchers have made use of CNN for different classification purposes like identifying lung diseases, segmenting brain tumors, and analyzing chest X-rays. Prior research has also looked at CNN for applications like text summarization, text sentiment analysis, and text report classification. CNN is being used to identify eye diseases that could potentially lead to vision loss through analysis of medical records. The proposed methodology for recognizing Fake news is depicted in Fig 2. The optimized CNN model in this research project has been tailored in terms of layer count, neuron count, and choice of optimizer. Table 3 contains specifics of the CNN architecture. Besides CNN, other popular ML models like RF, SGD, LR, and LSTM have also been utilized for comparison in this process. Fine-tuning various hyperparameters optimizes the performance of these models, with Table 3 providing a list of the specific parameters.

**Table 3. The learning models' layer structure and hyperparameters.**

| Classifier | Hyperparameter |
|---|---|
| CNN | Convolutional-layer with 7x7 kernel size and 64 filters. 2x2 size Max pooling and global max layer. Dropout rate of 0.5, dense layer with 32 neurons, softmax activation with 3 classes, and categorical-cross-entropy loss function. |
| LSTM | LSTM(200 neuron), Softmax (3), Dense(64 neuron), Dropout (0.5) |
| LR | default parameters |
| RF | number-of-trees = 150, random-state = 48, maximum-depth = 30 |
| VC | Voting criteria = soft |
| SGD | penality = "I2", loss = "log" |

## 3.4 Experiment models

On the selected dataset, this study evaluates how well the proposed method performs in comparison to many other ML and DL models. A summary of these models is given for thoroughness.

**3.4.1 Random Forest(RF).** RF classifier is one of the most promising ML techniques for detecting fake news [39]. To improve the model's overall classification accuracy and robustness, a lot of decision trees are combined using an ensemble learning technique called RF. Then, the RF algorithm creates multiple versions of decision trees from an individual subset of features on a random subset of input training data. For predicting, these trees will make their predictions at the final node. The final prediction in such an ensemble method would be determined by taking a majority vote or averaging out the projections of the individual trees. Several advantages can be derived from using RF for FND. First, RF's ensemble nature will enable it to identify intricate non-linear correlations between the target variable and the input characteristics, if characteristics of fake news are not easily identified through some simple rules or linear models. Second, it gives a measure of feature importance that helps find the most discriminative characteristics to ascertain differentiation between genuine and fake news, hence helping in further feature engineering and model interpretability [40]. Besides, RF trains many decision trees on random subsets of the data, so this shall be less prone to overfitting a single decision tree model and thus generalizes the model's performance on unseen data. Moreover, RF can operate with a wide variety of data: numerical, categorical, and textual features. Thus, it becomes versatile with the wide range of inputs usually found when performing tests to detect fake news. To classify news stories as authentic or fake, performance has been excellent while using an RF model with features that reflect textual, metadata, and context information.

**3.4.2 Logistic Regression(LR).** The other prominent ML algorithm within FND is logistic regression. LR can be conceptualized as a supervised learning approach, specially adapted for binary classification problems, such as classifying news stories as fake or genuine [41]. The Logistic Regression model estimates the probability of events based on input variables, predicting whether a news item is real or fake in this case. It learns coefficients that characterize the relationship of input variables with the target variable and uses these to make predictions about class real or fake on new unseen data. Some key reasons logistic regression can be used in FND involve its interpretable fashion since the coefficients of the model provide information on both the strength and direction of the association between the target variable and the input features, its ability to accommodate a wide range of features that include numerical, categorical, or even textual data, and lastly, probabilistic output, which might help set up threshold or confidence levels for appropriate classification. In addition, logistic regression models are typically resilient to highly connected cases of overfitting; they can also be huge with large

datasets [42]. Hence, they can be used in real-world applications which involve FND. Several researchers used Logistic Regression models trained on mixed features, including linguistic characteristics, source credibility, and social engagement patterns, and obtained promising results in grading news articles as real or fake. However, the thing to note here is that logistic regression can only be practical to a certain extent it might miss out on many complex, nonlinear relationships within the data in such scenarios, more advanced ML algorithms would draw a line of demarcation between nuanced patterns defining real news versus fake news.

**3.4.3 Stochastic Gradient Descent (SGD).** Besides the ML algorithms discussed above, optimization techniques such as SGD have also been used in FND [43]. SGD is one of the most popular optimization techniques used to tackle complex ML issues and has been utilized for phony news detection tasks, too. In contrast to stochastic gradient descent, which runs the optimization process based on one or a small subset of training examples at a time, batch gradient descent updates model parameters based on the entire training dataset. Among the advantages that SGD brings to FND is computational efficiency, as SGD requires fewer computations per iteration compared to batch gradient descent, especially when dealing with large datasets. It can also process streaming data, so it is pretty fit for FND systems in real-time. It is usually more robust to noisy or imbalanced training data, which often is the case in the context of FND. Another merit is that SGD is relatively simple and very easy to implement thus, it can be used within a wide range of machine-learning frameworks and architectures [44]. The researchers applied SGD-based optimization techniques to train a classifier against logistic regression and neural network models aimed at drawing almost imperceptible lines of distinction between genuine and fake news articles. Using SGD, models gracefully update their parameters against the evolving nature of fake news; this is important in maintaining accurate and up-to-date FND abilities. Performance with SGD-based methods may have a high sensitivity to such hyperparameters as the learning rate and batch size this requires careful tuning and monitoring of hyperparameters to enable the stability and convergence of the optimization process.

**3.4.4 LSTM.** Another interesting method for detecting fake news is the use of LSTM networks, a kind of RNN that excels at processing sequential input, like text [45]. Traditional RNN models could suffer from the vanishing gradient in the event of long-range dependencies. By contrast, LSTMs were designed to capture and memorize relevant information over more extended periods, thus able to model complex language patterns and contextual cues typical in news articles. In the domain of FND, LSTM models can be trained on textual content, metadata, and social engagement features of news articles to learn distinctive linguistic and behavioral features that distinguish real from fake news. The ability to model long-range dependencies allows LSTM-based models to learn complex interactions among parts of a news article and consider the general context in which it is shared and consumed. In the former case, the recurrent nature of LSTM models allows them for application in real-time or streaming scenarios, let through continuous monitoring and detection of rising fake news [46]. The methods that involved LSTMs showed an accuracy in opposition to traditional ML approaches, such as LR and SVM in classifying news articles into either natural or fake. The success of the LSTM network in the domain brings a message related to the use of advanced DL techniques to solve this new, complex, and evolving problem of FND.

## 3.5 Evaluation parameters

For FND this research work uses accuracy, F1 score, recall, and precision as evaluation metrics for all learning models' performance assessment. Accuracy is utilized for an overall percentage of correct predictions, be that for either natural or fake news [47]. It is determined by dividing

the number of accurate predictions by the total number of predictions.

$$Accuracy = \frac{TP + TN}{TP + TN + FP + FN} \tag{3}$$

Precision refers to how well your model detects fake news instances. The ratio of correctly identified phony news to the total optimistic predictions is called the actual positive rate [48].

$$Precision = \frac{TP}{TP + FP} \tag{4}$$

The recall is just another way of saying sensitivity and is used for the overall detection of fake news stances that the model correctly identified [49]. It is simply the ratio of the number of accurate fake news predictions in total instances of fake news.

$$Recall = \frac{TP}{TP + FN} \tag{5}$$

The precision and recall have combined into one metric, the F1-score.

$$F1 - Score = 2 \times \frac{Precision \times Recall}{Precision + Recall} \tag{6}$$

These evaluation metrics are critical to calculate the overall efficiency of ML models concerning FND. A practical model needs to balance F1-score, precision, accuracy, and recall so that issues with the effective detection of fake news will be less in terms of false positives and false negatives [50].

## 4 Experimental setup with results and discussion

The experiments are performed using Dell Optiplex T980 equipped with 8GB GPU and 64 GB of DDR5 Random Access Memory. The environment is Anaconda Jupyter-notebook with Python programming language. Sklearn, Keras, and Tensorflow are utilized for implementing ML and DL models.

### 4.1 Model predictive performance comparison using the FNC-1 dataset

Numerous tests have been conducted to detect fake news. Work is currently being done to create a reliable way to detect fake news. The experiments employ LSTM, CNN, RF, LR, and SGD as well as a voting classifier that blends two ML models. Various word-embedding methods, including GloVe, fastText, and fastText-subword, are being researched to detect fake news.

**4.1.1 Experiments using fastText.** The experimental results using all the learning models with fastText word-embedding are shown in Table 4. The CNN model outperforms the others with an accuracy of 90%, and it also shows strong precision (88%), recall (86%), and F1 score

**Table 4. Results of classifier classification using fastText.**

| Models | Accuracy(%) | Precision(%) | Recall(%) | F1-Score(%) |
|---|---|---|---|---|
| CNN | 90 | 88 | 86 | 87 |
| LSTM | 87 | 71 | 75 | 73 |
| RF | 87 | 80 | 87 | 82 |
| SGD | 87 | 77 | 88 | 82 |
| LR | 87 | 77 | 88 | 82 |
| VC (LR + SGD) | 87 | 77 | 88 | 82 |

**Table 5. Classifier classification outcomes using fastText-subword.**

| Models | Accuracy(%) | Precision(%) | Recall(%) | F1-Score(%) |
|---|---|---|---|---|
| CNN | 88 | 86 | 83 | 84 |
| LSTM | 88 | 85 | 81 | 82 |
| RF | 88 | 83 | 88 | 84 |
| SGD | 88 | 77 | 88 | 83 |
| LR | 88 | 77 | 88 | 82 |
| VC (LR + SGD) | 87 | 77 | 87 | 82 |

(87%). The LSTM model, while having a good accuracy of 87%, lags in precision (71%) and recall (75%), resulting in a lower F1 score of 73%. The RF model shows a slightly higher accuracy of 87%, with balanced precision (80%) and recall (87%), and an F1 score of 82%. Both the SGD and LR models have similar performances with accuracy close to 88% and identical precision (77%), recall (88%), and F1 scores (82%). The VC model, which combines LR and SGD, has an accuracy of 87.61% and maintains the same precision (77%), recall (88%), and F1 score (82%) as the individual SGD and LR models. Overall, CNN stands out as the best-performing model in terms of overall metrics.

**4.1.2 Experiments using fastText subword.** Table 5 provides a comparative analysis of the performance of different ML models using fastText with subword information. The CNN model shows an accuracy of 88%, 86% precision, 83% recall, and F1-score of 84%, indicating strong and balanced performance. Similarly, the LSTM model also achieves an accuracy of 88%, with a slightly lower precision of 85%, 81% recall, and 82% of F1-score. The RF model matches the accuracy of 88%, 83% precision, 88% high recall, and F1 score at 84%, showcasing its robustness in recall. The SGD and LR models both achieve an accuracy of 88%, 77% precision, 88% recall, and F1-scores of 83% and 82%, respectively. Lastly, the VC model, combining LR and SGD, has an accuracy of 87%, 77% precision, 87% recall, and 82% F1-score reflecting a slightly lower performance in accuracy while maintaining balanced precision and recall.

## 4.2 GloVe embedding experimental results

After that, FND is done using GloVe word-embedding. The results of all learning models are shared in Table 6. CNN model demonstrates strong performance with an accuracy of 87%, high precision of 89%, recall of 90%, and an F1 score of 89%, making it the best performer across most metrics. The LR model achieves an accuracy of 86%, with a precision of 77%, recall of 86%, and an F1 score of 80%, indicating decent but less balanced performance. The LSTM model has an accuracy of 85%, with precision at 79%, recall at 75%, and an F1 score of 77%, showing a notable drop in recall. The RF model also performs well with an accuracy of 87%,

**Table 6. Results of experiments utilizing the GloVe embeddings.**

| Models | Accuracy(%) | Precision(%) | Recall(%) | F1-Score(%) |
|---|---|---|---|---|
| CNN | 87 | 89 | 90 | 89 |
| LR | 86 | 77 | 86 | 80 |
| LSTM | 85 | 79 | 75 | 77 |
| RF | 87 | 82 | 87 | 81 |
| SGD | 87 | 74 | 87 | 80 |
| VC (LR + SGD) | 82 | 76 | 82 | 78 |

precision of 82%, recall of 87%, and an F1 score of 81%, suggesting strong recall and balanced metrics. The SGD model matches RF in accuracy at 87%, with a precision of 74%, recall of 87%, and an F1 score of 80%, highlighting a lower precision. Finally, the VC model, combining LR and SGD, achieves an accuracy of 82%, with precision of 76%, recall of 82%, and an F1 score of 78%, showing the lowest accuracy among the models but balanced precision and recall.

**4.2.1 Experimental results using combined word-embedding features.** Finally, to train the models, With proportions of 33%, 34%, and 33%, respectively, the three popular word-embedding techniques (fastText, fastText-subword, and Glove) features are merged. Next, important features are extracted by reducing dimensions through the use of PCA. It is noticeable that the performance of models is greatly enhanced by combining various word embedding techniques, as shown in Table 7. The CNN model exhibits exceptional performance with the highest accuracy of 94.58%, precision of 95.35%, recall of 97.29%, and F1 score of 96.11%, making it the top performer across all metrics. The LR model achieves an accuracy of 92.35%, with a precision of 91.11%, recall of 89.11%, and an F1 score of 90.11%, showing strong but slightly lower performance compared to CNN. The LSTM model has an accuracy of 90.35%, with a high precision of 92.47%, recall of 94.32%, and an F1 score of 93.39%, indicating very good performance, particularly in recall and F1 score. The RF model records an accuracy of 88.98%, precision of 90.11%, recall of 93.11%, and an F1 score of 91.16%, showing good overall performance with a focus on recall. The SGD model achieves an accuracy of 89.24%, with a precision and recall both at 92.11% and an F1 score of 92.11%, indicating balanced performance across metrics. Lastly, the VC model, combining LR and SGD, performs strongly with an accuracy of 93.65%, precision of 93.11%, recall of 94.11%, and an F1 score of 93.16%, showing robust and balanced performance, slightly outperforming individual LR and SGD models. The comparison of all learning models using combined features is shown in Fig 3.

**4.2.2 Comparison of performance of State-of-the-Art studies.** An analysis of performance is conducted in comparison to current research to show the importance of the proposed method. Teams that participated in the competition have been chosen for this specific reason. The comparative outcomes presented in Table 8 demonstrate that the proposed framework, which employs the CNN model with blended characteristics, performs better at identifying fake news than the most advanced algorithms.

## 4.3 Experiments using Arabic fake news dataset (AFND)

Besides utilizing the FNC-1 dataset, this research also incorporates the AFND dataset to confirm the effectiveness of the proposed method for identifying fake news. Tests were conducted using all the features for AFND and the outcomes are displayed in Table 9. The CNN model performs best overall, with its highest accuracy of 93.36% using Combined Features. The LSTM model also shows strong performance, achieving 93.00% accuracy with Combined

**Table 7. Experimental results of all learning models using the combined word-embedding features.**

| Models | Accuracy(%) | Precision(%) | Recall(%) | F1-Score(%) |
|---|---|---|---|---|
| CNN | 94.58 | 95.35 | 97.29 | 96.11 |
| LR | 92.35 | 91.11 | 89.11 | 90.11 |
| LSTM | 90.35 | 92.47 | 94.32 | 93.39 |
| RF | 88.98 | 90.11 | 93.11 | 91.16 |
| SGD | 89.24 | 92.11 | 92.11 | 92.11 |
| VC (LR + SGD) | 93.65 | 93.11 | 94.11 | 93.16 |

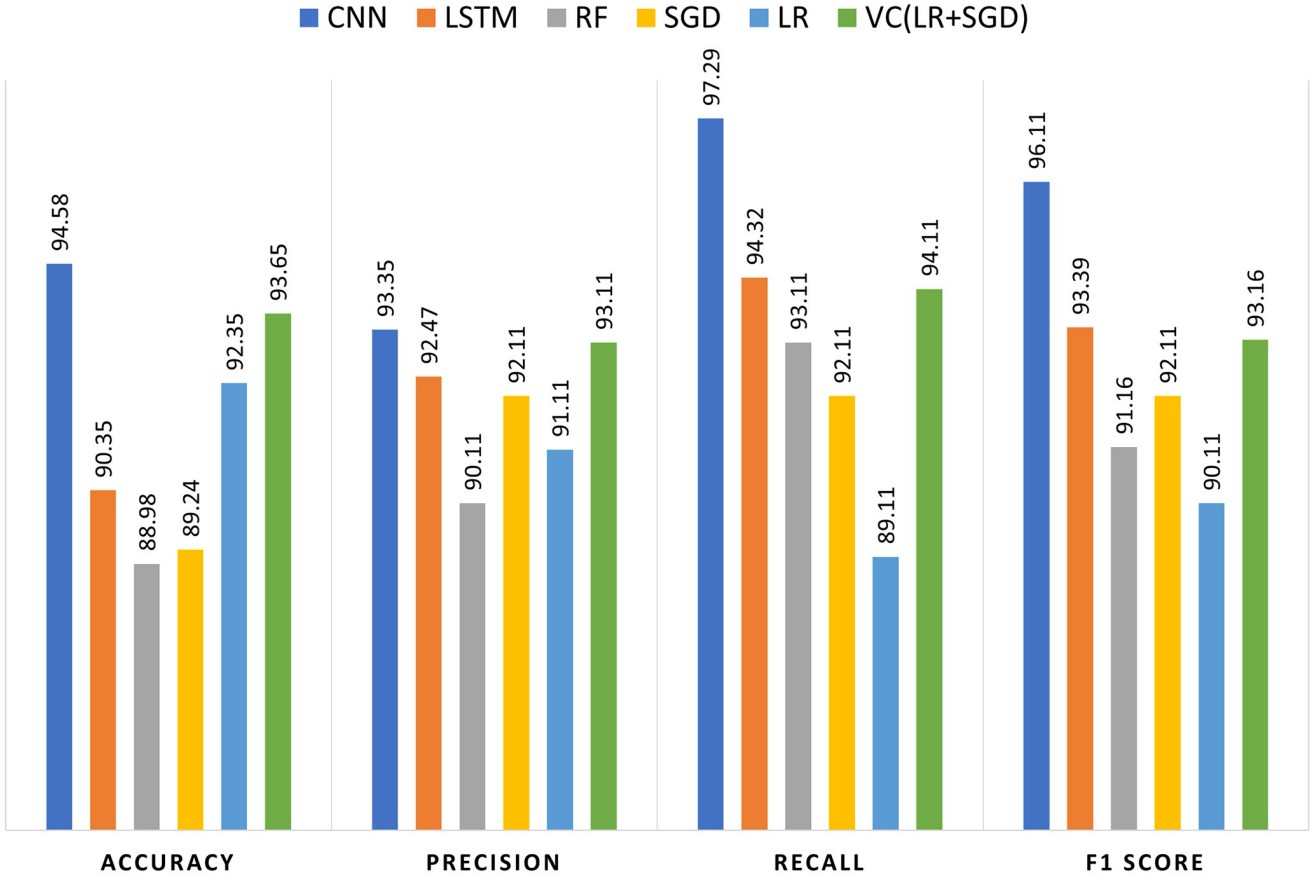

**Fig 3. Comparison of all learning models using combined features.**

Features. The RF model reaches 92.76%, while the VC model combining LR and SGD achieves 92.84% with Combined Features. The SGD and LR models show lower accuracies but improve with Combined Features, reaching 90.36% and 91.22%, respectively. The complete comparison graph is shown in Fig 4.

**Table 8. Comparison with previously published studies listed in research work [51].**

| Model | Accuracy Performance (%) |
|---|---|
| TalosComb | 82 |
| TalosTree | 83 |
| TalosCNN | 50.2 |
| Athene | 82 |
| UCLMR | 81.7 |
| FearMLP | 82.5 |
| StackLSTM | 82.1 |
| Upperbond | 85.9 |
| Proposed model | 94.58 |

**Table 9. Results of the experiment using every feature for dataset 2.**

| Models | fastText (%) | fastText-Subword (%) | GloVe (%) | Combined-Features (%) |
|---|---|---|---|---|
| CNN | 90.78 | 91.53 | 90.92 | 93.36 |
| LSTM | 89.99 | 90.74 | 92.22 | 93.00 |
| RF | 90.36 | 90.21 | 88.73 | 92.76 |
| SGD | 87.42 | 88.00 | 89.76 | 90.36 |
| LR | 87.35 | 88.32 | 88.34 | 91.22 |
| VC (LR + SGD) | 89.56 | 89.32 | 90.76 | 92.84 |

## 4.4 Significance of the proposed framework on on multi-lingual dataset

The TALLIP multi-lingual fake news detection dataset provides a unique opportunity to evaluate the effectiveness of fake news stance detection models across multiple languages, including English, Hindi, Indonesian, Swahili, Vietnamese, and a multilingual set [52]. The challenge of fake news detection across different languages lies in the variations in linguistic structure, contextual understanding, and cultural nuances. To tackle this issue, we proposed a novel approach leveraging a large word embedding blending technique integrated with a customized CNN model. The proposed model was rigorously evaluated on the TALLIP dataset, and the results were compared with several state-of-the-art models.

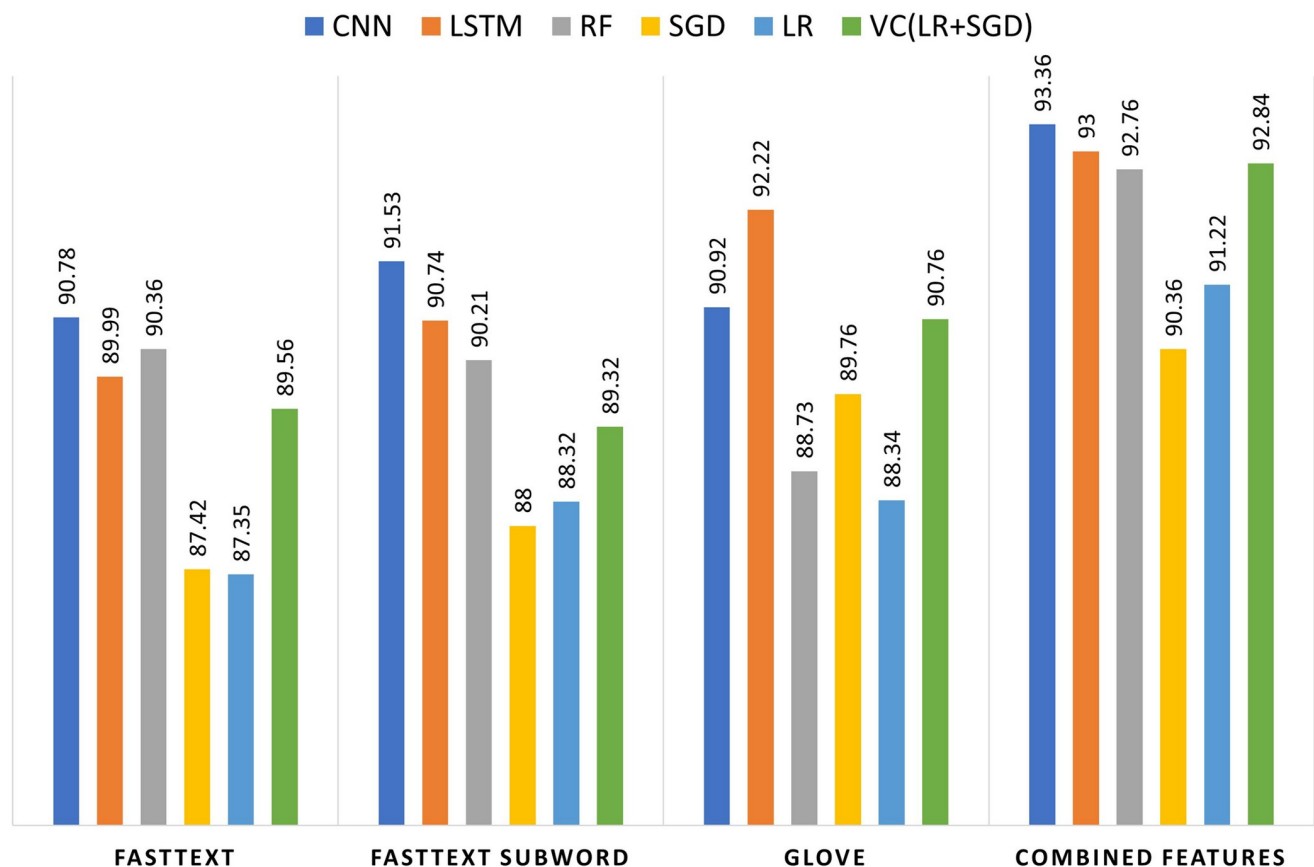

**Fig 4. Comparison graph of all word embedding with all learning models.**

**Table 10. Results comparison of SOTA models with respect to accuracy (%) on TALLIP multi-lingual dataset.**

| Framework | English | Hindi | Indonesian | Swahili | Vietnamese | Multilingual |
|---|---|---|---|---|---|---|
| **XLM-RoBERTa** [52] | 86.85 | 86.19 | 84.73 | 70.29 | 84.99 | 64.42 |
| **mBERT** [52] | 84.34 | 82.43 | 79.63 | 76.58 | 77.67 | 83.54 |
| **Semantic graph-based topic modelling** [53] | 88.58 | 86.31 | 88.09 | 83.99 | 83.18 | NA |
| **Bi-directional mBERT** [54] | 88.44 | 89.51 | 90.57 | 89.06 | 87.52 | 86.09 |
| **Proposed** | 89.15 | 90.19 | 90.95 | 90.76 | 90.34 | 89.64 |

Table 10 presents a comparison of the accuracy achieved by various models on the TALLIP dataset, including XLM-RoBERTa, mBERT, Semantic Graph-Based Topic Modelling, Bi-directional mBERT, and our proposed model. The results show that the proposed model consistently outperformed the SOTA models across all languages. Specifically, for English, the proposed model achieved an accuracy of 89.15%, which is higher than the best-performing Semantic Graph-Based Topic Modelling model (88.58%). In Hindi, the proposed model reached 90.19%, significantly outperforming Bi-directional mBERT (89.51%) and XLM-RoBERTa (86.19%). Similarly, for Indonesian and Swahili, the proposed model achieved accuracies of 90.95% and 90.76%, respectively, which is a substantial improvement compared to the baseline models, particularly outperforming XLM-RoBERTa, which struggled on Swahili with an accuracy of only 70.29%. For Vietnamese, our model also demonstrated superior performance with an accuracy of 90.34%, compared to the highest baseline of 87.52%.

The blending of FastText, FastText-Subword, and GloVe embeddings, combined with a customized CNN architecture, allowed the proposed model to capture nuanced semantic features from the text, contributing to its superior performance. The multi-embedding approach enables the model to handle variations in vocabulary and context, which is particularly crucial for low-resource languages like Swahili and Vietnamese. Additionally, by employing PCA for dimensionality reduction and selecting significant features, the proposed model efficiently managed the complexity of the dataset, preventing overfitting and ensuring generalizability across different languages and data splits.

## 5 Conclusion

This research presents a novel approach for detecting fake news by focusing on the correlation between the headline and the body of the news article, diverging from previous studies that predominantly analyzed isolated statements or phrases. With the exponential increase in manipulated content across digital platforms, there is an even greater need for an automated system to detect fake news at an early stage and with a high accuracy rate. The ensemble model proposes popular word-embedding techniques(FastText, FastText-Subword, and GloVe) combinational convolutional neural network conceived to improve feature extraction and accuracy in classification. Principal Component Analysis simplifies high-dimensional word vectors by retaining only the core features, thereby enhancing model efficiency. We conducted experiments using the Fake News Challenge dataset, and our model returned very impressive performance metrics: 94.58% accuracy, 95.35% precision, and 97.29% recall, with a 96.11% F1 score. Its robustness and superiority were further validated through various comparisons with ML, DL, and ensemble learning models, along with testing on an independent Arabic Fake News dataset. The study thus underlines the potential of our model to be a significant mitigation tool against the ill effects of fake news, offering open-source, scalable, and reliable solutions for real-time news verification in digital platforms. Our model innovatively integrates multiword embedding techniques and PCA to provide a blanket approach to the challenge of FND. The

future work direction for FND is the fusion of hand-crafted and word-embedding features with transfer learning models.

## Supporting information

**S1 File. The implementation code of this research work is shared as a supporting file "S1 File.ipynb".**
(IPYNB)

## Author Contributions

**Conceptualization:** Abdulaziz Altamimi.

**Data curation:** Abdulaziz Altamimi.

**Formal analysis:** Abdulaziz Altamimi.

**Funding acquisition:** Abdulaziz Altamimi.

**Investigation:** Abdulaziz Altamimi.

**Methodology:** Abdulaziz Altamimi.

**Project administration:** Abdulaziz Altamimi.

**Resources:** Abdulaziz Altamimi.

**Software:** Abdulaziz Altamimi.

**Supervision:** Abdulaziz Altamimi.

**Validation:** Abdulaziz Altamimi.

**Visualization:** Abdulaziz Altamimi.

**Writing – original draft:** Abdulaziz Altamimi.

**Writing – review & editing:** Abdulaziz Altamimi.

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
