## [Decision Letter · Decision Letter 0]

5 Aug 2024

PONE-D-24-26648Novel Approach For Predicting Fake News Stance Detection Using Large Word Embedding Blending and Customized CNN ModelPLOS ONE

Dear Dr. Altamimi,

Thank you for submitting your manuscript to PLOS ONE. After careful consideration, we feel that it has merit but does not fully meet PLOS ONE’s publication criteria as it currently stands. Therefore, we invite you to submit a revised version of the manuscript that addresses the points raised during the review process.

We look forward to receiving your revised manuscript.

Kind regards,

Lipeng Song

Academic Editor

PLOS ONE

Journal Requirements:

Please ensure that your manuscript meets PLOS ONE's style requirements, including those for file naming. The PLOS ONE style templates can be found at https://journals.plos.org/plosone/s/file?id=wjVg/PLOSOne_formatting_sample_main_body.pdf and https://journals.plos.org/plosone/s/file?id=ba62/PLOSOne_formatting_sample_title_authors_affiliations.pdf 2. Please note that PLOS ONE has specific guidelines on code sharing for submissions in which author-generated code underpins the findings in the manuscript. In these cases, we expect all author-generated code to be made available without restrictions upon publication of the work. Please review our guidelines at https://journals.plos.org/plosone/s/materials-and-software-sharing#loc-sharing-code and ensure that your code is shared in a way that follows best practice and facilitates reproducibility and reuse.
 3. Thank you for stating the following in your Competing Interests section:  "No" Please complete your Competing Interests on the online submission form to state any Competing Interests. If you have no competing interests, please state "The authors have declared that no competing interests exist.", as detailed online in our guide for authors at http://journals.plos.org/plosone/s/submit-now  This information should be included in your cover letter; we will change the online submission form on your behalf. 4. When completing the data availability statement of the submission form, you indicated that you will make your data available on acceptance. We strongly recommend all authors decide on a data sharing plan before acceptance, as the process can be lengthy and hold up publication timelines. Please note that, though access restrictions are acceptable now, your entire data will need to be made freely accessible if your manuscript is accepted for publication. This policy applies to all data except where public deposition would breach compliance with the protocol approved by your research ethics board. If you are unable to adhere to our open data policy, please kindly revise your statement to explain your reasoning and we will seek the editor's input on an exemption. Please be assured that, once you have provided your new statement, the assessment of your exemption will not hold up the peer review process. 5. Please review your reference list to ensure that it is complete and correct. If you have cited papers that have been retracted, please include the rationale for doing so in the manuscript text, or remove these references and replace them with relevant current references. Any changes to the reference list should be mentioned in the rebuttal letter that accompanies your revised manuscript. If you need to cite a retracted article, indicate the article’s retracted status in the References list and also include a citation and full reference for the retraction notice.

Reviewers' comments:

Reviewer's Responses to Questions

**Comments to the Author**

1. Is the manuscript technically sound, and do the data support the conclusions?

Reviewer #1: Yes

2. Has the statistical analysis been performed appropriately and rigorously? 

Reviewer #1: N/A

3. Have the authors made all data underlying the findings in their manuscript fully available?

Reviewer #1: Yes

4. Is the manuscript presented in an intelligible fashion and written in standard English?

Reviewer #1: Yes

5. Review Comments to the Author

**Reviewer #1:** In this manuscript “Novel Approach For Predicting Fake News Stance Detection Using Large Word Embedding Blending and Customized CNN Model”, in response to the rampant phenomenon of fake news, the author propose a novel method to automatically detect and identify fake news. This method is a combination of three popular word embedding techniques and integrated with a customized convolutional neural network. Moreover, this method’s effectiveness is validated on an independent Arabic Fake News dataset. Given the valuable method provided by this research, I recommend the manuscript for publication in PLOS ONE, with the suggestion to address the following comments to further refinement.

Main comments:

Comment 1: The novel method proposed by the author combines three popular word embedding techniques. Can you describe in detail how these three techniques are combined, and present this combined way in a diagram.

Comment 2: The authors should pay attention to the standardization of symbols, such as: 1) in subsection “3.3.3 fastText”, ”for” and ”give”, should be “for” and “give”; 2) in subsection “3.3.8 Modeling Methods”, variables y, i should be italic.

Comment 3: In subsection 4.1, comparing the experimental results of different methods, it is recommended to present them in figures for a more intuitive way.

6. PLOS authors have the option to publish the peer review history of their article (what does this mean?). If published, this will include your full peer review and any attached files.

Reviewer #1: No

---

## [Author Response · Author response to Decision Letter 0]

8 Aug 2024

I have provided a separate PDF for reviewers response.

---

## [Decision Letter · Decision Letter 1]

27 Sep 2024

PONE-D-24-26648R1Novel Approach For Predicting Fake News Stance Detection Using Large Word Embedding Blending and Customized CNN ModelPLOS ONE

Dear Dr. Altamimi,

Thank you for submitting your manuscript to PLOS ONE. After careful consideration, we feel that it has merit but does not fully meet PLOS ONE’s publication criteria as it currently stands. Therefore, we invite you to submit a revised version of the manuscript that addresses the points raised during the review process.

We look forward to receiving your revised manuscript.

Kind regards,

Jawad Rasheed, Ph.D.

Academic Editor

PLOS ONE

Reviewers' comments:

Reviewer's Responses to Questions

**Comments to the Author**

1. If the authors have adequately addressed your comments raised in a previous round of review and you feel that this manuscript is now acceptable for publication, you may indicate that here to bypass the “Comments to the Author” section, enter your conflict of interest statement in the “Confidential to Editor” section, and submit your "Accept" recommendation.

Reviewer #2: (No Response)

Reviewer #3: (No Response)

2. Is the manuscript technically sound, and do the data support the conclusions?

Reviewer #2: Partly

Reviewer #3: Yes

3. Has the statistical analysis been performed appropriately and rigorously? 

Reviewer #2: N/A

Reviewer #3: Yes

4. Have the authors made all data underlying the findings in their manuscript fully available?

Reviewer #2: Yes

Reviewer #3: No

5. Is the manuscript presented in an intelligible fashion and written in standard English?

Reviewer #2: No

Reviewer #3: Yes

6. Review Comments to the Author

Reviewer #2: 1) The use of standard word embeddings like FastText, FastText-Subword, and GloVe, combined with CNN, is well-explored in the field of NLP. This research does not present any significant innovation in terms of methodology or model architecture.

2) The explanation of how the CNN model is customized is vague. The current version lacks a clear description of the specific innovations in the CNN architecture, making it difficult to assess its contribution.

3)The model's accuracy and other performance metrics are reported, there is no discussion on the practical impact of the model, such as how well it performs under real-world conditions where data can be noisier and more complex.

4)The Fake News Challenge dataset is commonly used but may not fully represent the complexity of fake news detection. No details are provided on the characteristics of the Arabic Fake News dataset, raising questions about the robustness of cross-lingual performance claims.

5)The use of PCA for feature management is mentioned but not justified. How PCA contributes to the performance improvement or how much dimensionality was reduced is left unexplained, making it unclear whether this step was necessary.

6) The current version suggests the framework shows "superior performance" when compared with other approaches, but no substantial evidence is presented to support these claims beyond metrics. There's no detailed comparison with alternative models or baselines.

7) While the model reportedly performs well on an Arabic dataset, there is no detailed validation process to show how it generalizes across different languages and news genres, making the cross-lingual claim questionable.

Reviewer #3: 1. The language of the paper should be improved. Tools like Grammarly can be helpful. Some example statements that have punctuation and grammar mistakes:

a. The pernicious impact of such misinformation [2]. hits all sectors of society.

b. This study uses ML models for the accurate classification of fake news stances on social media. the main contribution of the study is ;

c. This stage in the process of processing natural language is crucial.

2. Table 4 metrics are not clear. The column titles can be modified as below:

Accuracy (%) Precision (%) Recall (%) F1-Score (%)

3. For Table 5, 6,7, 8, and 9 the % characters can be moved to the column titles. That would improve the readability of the tables. For example, the column titles of Table 9 can be as below:

Models fastText (%) fastText-Subword (%) GloVe (%) Combined-Features (%)

Then, the % signs next to the result numbers can be removed.

4. To improve the contribution to future research, the used code can be uploaded to a platform like GitHub, and a link can be provided in the paper.

7. PLOS authors have the option to publish the peer review history of their article (what does this mean?). If published, this will include your full peer review and any attached files.

Reviewer #2: No

Reviewer #3: **Yes: **Ahmet Gürhanlı

---

## [Author Response · Author response to Decision Letter 1]

16 Oct 2024

I have provided a separate PDF to address reviewers' concerns.

---

## [Decision Letter · Decision Letter 2]

6 Nov 2024

Novel Approach For Predicting Fake News Stance Detection Using Large Word Embedding Blending and Customized CNN Model

PONE-D-24-26648R2

Dear Dr. Altamimi,

We’re pleased to inform you that your manuscript has been judged scientifically suitable for publication and will be formally accepted for publication once it meets all outstanding technical requirements.

Kind regards,

Academic Editor

PLOS ONE

Additional Editor Comments (optional):

Reviewers' comments:

Reviewer's Responses to Questions

**Comments to the Author**

1. If the authors have adequately addressed your comments raised in a previous round of review and you feel that this manuscript is now acceptable for publication, you may indicate that here to bypass the “Comments to the Author” section, enter your conflict of interest statement in the “Confidential to Editor” section, and submit your "Accept" recommendation.

Reviewer #2: All comments have been addressed

Reviewer #3: All comments have been addressed

2. Is the manuscript technically sound, and do the data support the conclusions?

Reviewer #2: Yes

Reviewer #3: Yes

3. Has the statistical analysis been performed appropriately and rigorously? 

Reviewer #2: Yes

Reviewer #3: Yes

4. Have the authors made all data underlying the findings in their manuscript fully available?

Reviewer #2: Yes

Reviewer #3: Yes

5. Is the manuscript presented in an intelligible fashion and written in standard English?

Reviewer #2: Yes

Reviewer #3: Yes

6. Review Comments to the Author

Reviewer #2: Accept in Current Form , Authors Addressed all comments and Paper now can ve considered for publication.

Reviewer #3: (No Response)

7. PLOS authors have the option to publish the peer review history of their article (what does this mean?). If published, this will include your full peer review and any attached files.

Reviewer #2: No

Reviewer #3: No

---

## [Editor Report · Acceptance letter]

15 Nov 2024

PONE-D-24-26648R2 

PLOS ONE

Dear Dr. Altamimi, 

I'm pleased to inform you that your manuscript has been deemed suitable for publication in PLOS ONE. Congratulations! Your manuscript is now being handed over to our production team.

Kind regards, 

on behalf of

Dr. Jawad Rasheed 

Academic Editor

PLOS ONE